# Biliary Atresia: A Complex Hepatobiliary Disease with Variable Gene Involvement, Diagnostic Procedures, and Prognosis

**DOI:** 10.3390/diagnostics12020330

**Published:** 2022-01-27

**Authors:** Consolato M. Sergi, Susan Gilmour

**Affiliations:** 1Stollery Children’s Hospital, Laboratory Medicine and Pathology, University Alberta Hospital, University of Alberta, Edmonton, AB T6G 2B7, Canada; 2Division of Anatomic Pathology, Children’s Hospital of Eastern Ontario, Ottawa, ON K1H 8L1, Canada; 3Department of Pediatric Gastroenterology and Nutrition, University of Alberta, Edmonton, AB T6G 2B7, Canada; gilmour@ualberta.ca

**Keywords:** biliary atresia, cholestasis, newborn, genetics, virus

## Abstract

The diagnosis of biliary atresia is still terrifying at the 3rd decade of the 21st century. In a department of neonatal intensive care unit, parents and physicians face a challenge with a jaundiced baby, who may or may not have a surgically correctable hepatopathy. The approach has been systematically evaluated, but the etiology remains ambiguous. The study of families with recurrent biliary atresia has been undertaken at a molecular level. The primary interest with this disease is to identify the etiology and change the treatment from symptomatic to curative. The occurrence of this obstructive cholangio-hepatopathy in well-known genetic syndromes has suggested just coincidental finding, but the reality can be more intriguing because some of these diseases may have some interaction with the development of the intrahepatic biliary system. Several genes have been investigated thoroughly, including *ADD3* and *GPC1* shifting the interest from viruses to genetics. In this review, the intriguing complexities of this hepatobiliary disease are highlighted.

## 1. Introduction

The term biliary atresia has substituted the original term of “extrahepatic biliary atresia”, which has been in use for several generations. The concept was related to the often-identified absence of gallbladder with an obliterated cord at the site the extrahepatic biliary system. It is now known that biliary atresia is a necro-inflammatory and fibro-obliterative process of both intrahepatic and extrahepatic biliary tract [1]. Despite the fact that the etiology is far from being known, pediatric surgeries conducted last century worldwide were incredibly supportive for the clarification of the nosology of this disease [2,3,4]. Mainly, Morio Kasai, a Japanese pediatric surgeon, may be considered the pioneer of a specific pediatric surgery that may be offered to parents of a child with biliary atresia. His name is indelibly bound to the palliative surgical procedure he ideated in Japan [5]. The porto-enterostomy is indeed the major palliative procedure, which may be implemented before a liver is available for transplantation (orthotopic transplant or transplant of a liver from a recently deceased donor, a living donor transplant, or a split type of liver transplant) can be put forward. The biliary atresia is a very complex disease, not just complicated, and several genes have been investigated thoroughly. Some of these genes may be highly relevant to this disease, although their role is mostly not known. Currently, epigenetics is playing a significant role in several hepatobiliary diseases, and we may predict it may play a significant role in biliary atresia, particularly when the genetic platforms have been delineated in more detail [6]. Epigenetic events may occur at the end of the third trimester of pregnancy and may consequently act on a hepato-biliary disease. In this review, we focus on the intriguing complexities of this hepatobiliary disease.

## 2. Materials and Methods

This review is based on the 30-years personal experience of patients with biliary atresia and literature searches performed in PubMed, Scopus, and Google Scholar on “biliary atresia” of the last 23 years (1998–2021) using “biliary atresia“, “genetics“, and “diagnosis“ as key words. A formal analysis of single cases with biliary atresia would expand the study in two continents for both authors, but such a review of single cases would probably be better labeled as clinical study. Thus, the authors opted to expand this systematic review instead. They concentrated on their personal experience and re-evaluated anonymous microphotographic material of patients affected with biliary atresia encountered during their 30-years professional activity.

## 3. Histology of the Developing Fetal Liver

One of the questions that we usually receive from students and fellows is why the study of the development of the fetal liver should be valuable in the study and management of the biliary atresia. Although controversially discussed, and probably interpreted, liver ductal plate remnants are probably essential for the prognosis of patients affected with biliary atresia as shown by us and others [7,8,9,10,11,12,13,14,15,16]. During gastrulation, the endoderm germ layer is set. It forms a primitive gut tube that is subdivided into foregut, midgut, and hindgut regions [17,18,19]. In the mouse embryo at embryonic day 8.0 of gestation, the embryonic liver buds from the ventral foregut endoderm [20]. The hepatic diverticulum forms at first. The thickened ventral foregut epithelium outpouring from the foregut is adjacent to the developing heart at embryonic day 9.0, and the anterior portion of the hepatic diverticulum gives rise to the liver and intrahepatic biliary tree, while the posterior portion establishes the gallbladder and extrahepatic bile ducts [21]. The liver bud occurs at embryonic day 9.5, when the hepatic endoderm cells, also called as hepatoblasts, delaminate from the epithelium and invade the adjacent septum transversum mesenchyme (STM). The fibroblasts and stellate cells of the liver originate from the STM. In the embryonic days between the 10th and the 15th, the liver bud is vascularized and colonized by hematopoietic cells and becomes the major fetal hematopoietic organ. The bipotential hepatoblasts that reside next to the portal veins become biliary epithelial cells or cholangiocytes. These cells will line the lumen of the intrahepatic bile ducts (IHBD), while the majority of hepatoblasts in the parenchyma differentiate into hepatocytes [22,23]. The maturation of functional hepatocytes and the formation of a biliary network connected to the extrahepatic bile ducts (EHBD) are gradual. In humans, this process has been delineated previously [22,24]. From 12 weeks of gestation on, a progressive remodeling of the ductal plate, which is the protostructure of the human biliary system, occurs. In a few weeks, a few parts of the primitive biliary structure, which are the peripheral tubular or ductular structures progressively dilate. Then, they slowly migrate toward the center of the portal tract. The centripetal migration is associated with their transformation into mature bile ducts. During the centripetal migration and transformation, most of them gradually disappear. Thus, the primitive fetal biliary structure or a ductal plate of the liver is remodeled during intrauterine life into the biliary drainage system that is going to be used for all life. This process of selection and deletion is ordered, but little is known which part is going to be deleted and which transform into the mature interlobular bile duct despite the apoptosis playing a significant role as a selective mechanism as identified almost two decades ago [25]. Previously, our morphometric data indicate that a ‘slow-down’ period of the progressive ramification of the intrahepatic biliary tree occurs between the 20th week and the 32nd week of gestation [22]. The growth of the intrahepatic portal tracts, described in terms of cross-section circumference and the encircled area, shows a progressive deceleration indeed, i.e., between the end of the 2nd trimester and the beginning of the 3rd trimester.

Conversely, the development of bile ducts had an acceleration of the process after the 32nd week of gestation and before 20 weeks of gestation. We found out that the interim hematopoietic function of the liver and, specifically, the intraportal granulopoiesis may play a role. Remarkably, the necro-inflammatory and fibro-obliterative process of biliary atresia may be associated with both an abnormal ductal plate development and extramedullary hematopoiesis. With regard to the intrauterine hematopoiesis, liver erythropoiesis leads between 12 weeks of gestation until the beginning of the 3rd trimester of pregnancy, specifically the 25th week of gestation. At this time, about half of the blood cells are formed in the liver and half in the bone marrow. It is, however, at 32 weeks of gestation that the liver plays a major role in both the hematopoiesis and the materno-fetal exchange. In the past, investigations carried out on our collectives of specimens revealed only at this time that hematopoietic cells in the liver begin to form islands out of a previously diffuse distribution. It is only after birth that the bone marrow becomes the major site of production of both red and white cells series.

## 4. Pathological Anatomy of the Biliary Atresia

The histology of the liver with biliary atresia is variable and depends on the stage when the biopsy is done substantially. At the early stage, fibrosis and biliary ductular proliferation, two main essential criteria, are present but may be very focal and missed at first glance. A subsequent liver biopsy may harbor more conclusive findings. However, the liver biopsy is essential to indicate if the main inflammatory component is at the level of the portal tracts or at acinar level, which may indicate neonatal hepatitis [7,14]. Thus, in biliary atresia, there are prominent histologic features, including a limited involvement of the liver acinus with very few multinucleated giant hepatocytes, which should have six or more nuclei. The limitation of acinus involvement is critical. At the late stage, fibrosis becomes more prominent, and the chances to be successful with a hepatic portoenterostomy are limited [6,26]. At the late stage, liver cirrhosis is usually encountered with pseudolobules and lobules surrounded by expanded portal tracts with biliary ductular proliferation (Figure 1). 

One of the most detailed studies on biliary atresia was performed at the Division of Surgery, Children’s Research Hospital, Kyoto Prefectural University of Medicine, Kyoto, Japan. In this clinic-pathological investigation, 31 patients with uncorrectable obstructive cholangiopathy underwent hepatic portoenterostomy and steroid therapy between 1988 and 2005 [16]. An immunohistochemical investigation was performed using an antibody against keratin 19 of the intermediate filament family of the cytoskeleton. In this study, ductal plate malformation (DPM) typing was performed according to generally accepted criteria [27]. Only type II DPM was considered relevant, and specimens were deemed as DPM-positive if a concentric cellular arrangement was detected. Shimadera et al. found that the presence of DPM in the liver of patients with biliary atresia predicts poor bile flow after hepatoportoenterostomy [16]. This study was retrospective and included comparisons of preoperative characteristics, the postoperative jaundice period, and cumulative steroid doses between patients with and without DPM. Despite numerous reports on the DPM of biliary atresia, the influence of biliary remnants remains controversially discussed in the literature. During the re-examination of our microphotographic material, we identified abnormalities, such as cytomegalovirus expression in the placenta (Figure 2). Visualization of cytokeratins (kerains) and neural cell adhesion molecules in the immature ductal cells by means of the immunohistochemical method can be a useful tool for the microscopic examination of the immature biliary structures in the liver. Russo et al. [28] assessed the relative value of histologic features in 227 liver needle biopsies in discriminating between biliary atresia and other cholestatic disorders in infants enrolled in a prospective Childhood Liver Disease Research and Education Network (ChiLDReN) cohort study. These authors correlated histology with clinical findings in infants with and without biliary atresia. Also, Russo et al. reviewed 316 liver biopsies from clinically proven biliary atresia patients and correlated histologic features with total serum bilirubin 6 months post-Kasai or hepatoportoenterostomy and transplant-free survival up to 6 years. Logistic regression analysis determined that bile plugs in portal bile ducts/ductules, moderate to marked ductular reaction and portal stromal edema had the largest odds ratio for predicting biliary atresia vs. non- biliary atresia. Remarkably, the diagnostic accuracy of the needle liver biopsy was assessed to be 90.1%, whereas sensitivity and specificity for a diagnosis of biliary atresia were 88.4% and 92.7%, respectively. Russo et al. found that higher stages of fibrosis, a ductal plate configuration, moderate to marked bile duct injury, an older age at hepatoportoenterostomy and an elevated INR (international normalized ratio), which is a type of calculation based on prothrombin test results, were independently associated with a higher risk of transplantation [28].

Apart from biliary atresia major causes of cholestatic liver disease in infants include choledochal cyst, Caroli’s syndrome, Alagille syndrome, and neonatal infection (Cytomegalovirus Herpesvirus Hepatotropic viruses, Human immunodeficiency virus, Parvovirus B19, Paramyxovirus, Enteric viruses, Rubella, Bacterial sepsis, Listeriosis, Toxoplasmosis, Syphilis). Also, disorders of carbohydrate metabolism, disorders of amino acid metabolism, disorders of glycolipid and lipid metabolism, disorders of glycoprotein metabolism, metal storage disorders, peroxisomal disorders, mitochondrial cytopathies, hereditary disorders of bilirubin metabolism, hereditary disorders of bile formation, disorders of bile acid biosynthesis, disorders of protein biosynthesis and targeting (e.g., α1-Antitrypsin deficiency) and miscellaneous disorders (e.g., Aagenaes syndrome, citrullinemia, type II X-linked adrenoleukodystrophy, shock/hypoperfusion, parenteral nutrition, fetal alcohol syndrome, drugs, Budd-Chiari syndrome, multiple hemangiomas) as well as neoplastic diseases (e.g., neonatal leukemia, neuroblastoma, hepatoblastoma, Langerhans cell histiocytosis, and erythrophagocytic lymphohistiocytosis). Although the discussion in detail of each disorder will be the topic of a future study and may be outside of the scope of this current review, a few comments need to be added. The broad differential diagnosis of jaundice in infants and children presupposes a good clinical and pathological correlation and periodic liver rounds. Since the evaluation of pediatric liver biopsies is quite distinct from that of adults, the liver pathologist needs to possess a good knowledge of biochemistry and metabolic disorders as well as dysmorphology understanding. The performance of all potentially necessary tests when a liver biopsy is carried out, prior arrangements should be in place between the clinician, the radiologist, and the laboratory to ensure adequate specimen processing and rapid workout. The laboratory should have liquid nitrogen on site or, better, at the bedside. Cores of tissue, up to 2 cm in length, should be snap frozen in liquid nitrogen refrigerated isopentane immediately to preserve tissue integrity and mRNA (messenger ribonucleic acid). The tissue fragments should then be placed within a specimen vial and a small portion of tissue may be placed in glutaraldehyde for electron microscopy processing, and a portion can be sent to the biochemistry laboratory for specific requests. 

## 5. Genetic Complexities and Perspectives

The development of omics technologies has shown impressive speed in the last two decades, and the application of Genomic Wide Associated Study (GWAS) has evidenced that we may concentrate on two major genes, including *ADD3* and *GPC1* [29,30,31,32]. *ADD3* is a gene, located in the 10q24.2 region, which has been indicated as a susceptibility gene for biliary atresia [33,34]. Zeng et al. studied two single nucleotide polymorphisms (SNPs) in the *ADD3* gene, rs17095355 C/T and rs10509906 G/C, in a Chinese population [34]. These authors indicated that the rs17095355 SNP was significantly associated with an enhanced risk to develop biliary atresia under the genotypic, dominant, recessive, and additive models. On the other hand, the other SNP, rs10509906, only presented a significant protective effect under the additive model. Zeng et al.’s haplotype analysis suggested that these two SNPs had a certain interaction within a haplotype to influence the risk to develop biliary atresia. In particular, the T_a_-G_b_ haplotype was associated with an increased risk to develop biliary atresia compared with the C_a_-C_b_ haplotype [34]. A recent meta-analysis confirmed the extraordinary importance of the rs17095355 SNP in Asian populations [35]. These authors investigated five relevant studies involving 1000 patients and 3257 controls to analyze the association between rs17095355 and risk to develop biliary atresia. The pooled odds ratio for T allele of rs17095355 was 1.72 (95%CI 1.53–1.92, *p* < 0.01) in patients with biliary atresia. Stratification by ethnicity indicated the degree of risk of rs17095355 with susceptibility for biliary atresia was similar in populations of Asian origin. The pooled odds ratio was detected at 1.81 (95%CI 1.60–2.06, *p* < 0.01). It is important to emphasize that more original studies with the large sample are needed to replicate this genetic association in different ethnics. 

Through the GWAS, an additional putative gene, which is called glypican 1 (*GPC1*), was identified. The *GPC1* gene is located on chromosome 2q37.3. Glypicans are heparan sulfate proteoglycans. Glypicans are bound to the external surface of the plasmatic membrane of a cell by a glycosyl-phosphatidylinositol (GPI) linkage [30,36]. Homologs of glypican molecules have been identified throughout the *Eumetazoa*. On the other hand, clear glypican homologs are not found outside the Metazoa. The family of these proteoglycans includes six members (GPC1 to GPC6). Glypican family members are tangled in numerous pathways of signaling and developmental significance regarding the hepatocytes and cholangiocytes. Moreover, there is the role of GPC-3 for being a marker and a therapeutic target of hepatocellular carcinoma [37]. In 2018, Sangkhathat et al. used a whole-exome sequencing approach to look for other cholestasis entities in twenty cases diagnosed with biliary atresia in Thailand [38]. These authors targeted nineteen genes associated with infantile cholestasis syndromes. Variant selection focused on those with allele frequencies in dbSNP150 database of less than 0.01. A polymerase chain reaction (PCR)-direct sequencing was used to verify all selected variants. Of the 20 cases studied, thirteen rare variants were detected in nine genes: four in *JAG1* (Alagille syndrome), two in *MYO5B* (progressive familial intrahepatic cholestasis [PFIC] type 6), and one each in *ABCB11* (PFIC type 2), *ABCC2* (Dubin-Johnson syndrome), *ERCC4* (Fanconi anemia), *KCNH1* (Zimmermann-Laband syndrome), *MLL2* (Kabuki syndrome), *RFX6* (Mitchell-Riley syndrome), and *UG1A1* (Crigler-Najjar syndrome). The authors concluded that a severe inflammatory cholangiopathy in biliary atresia might be a shared pathology among several infantile cholestatic syndromes, as demonstrated previously [7]. Although these results may be controversially discussed, the omics studies open a Pandora’s box to find out numerous other genes that may be correlated with biliary atresia at some point of the natural history of this disease. Thus, far from being univocal, the genetic research on biliary atresia revealed that other genes involved in biliary tract dysmorphogenesis and cholestasis, the immunologic response, vasculogenesis, and left-right patterning might contribute to this disease with worrisome complications and prognosis. Currently, clinical investigations and nonhuman model systems orientate also on *CFC1*, *CFTR*, *JAG1*, *IFN-γ*, *INV*, *MIF*, *VEGF*, *SOX17*, and *ZIC3* [6,33,38,39,40,41,42,43,44,45,46,47,48,49,50,51,52,53,54,55,56,57,58,59,60]. 

Next-generation sequencing (NGS) has been a pillar in identifying several variants of channelopathies in sudden cardiac death, and its use has spread-out on numerous fields of medicine, particularly cardiology and gastroenterology [61,62,63]. NGS includes the implementation of miniaturized and parallelized sequencing platforms targeting from 1 million to 43 billion short reads (50–400 bases each) per instrument run. Different from the polymerase chain reaction-based Sanger sequencing, NGS runs numerous parallel sequences. In the PCR-based Sanger sequencing, the separation of chain-termination products performed on polyacrylamide gel electrophoretic had an individual character only. However, NGS requires sophisticated bioinformatics systems, high-speed data processing, and extensive data storage capabilities, which can be costly without adequate bioinformatic support [6].

You et al. aimed to develop a genetic method to investigate the pathogenesis of biliary atresia in mice with experimental biliary atresia [64]. These authors targeted the gene expression profile of biliary atresia using the Gene Expression Omnibus database and included 18 samples from newborn mice. The samples were collected at three time points following the induction of an experimental biliary atresia. The differentially expressed genes (DEGs) in mice with biliary atresia were identified using the limma package in R language and hierarchical clustering analysis. R is a software language for statistical computing and graphics. The R Core Team and the R Foundation support it for Statistical Computing. You et al. identified 306 DEGs in the samples from the 3 day time point, 721 at 7 days and 370 at 14 days. Seventy-four common DEGs were identified in these three sample groups, which are reported to function in multiple immune biological processes, including the defense response, leukocyte migration, cell chemotaxis and leukocyte chemotaxis. Six common DEGs (CCL3, CXCL5, CXCL13, CXCR2, CCL5 and CCL6) were identified that were involved in the significantly enriched functions and the significantly enriched pathways. Of note, CCL3 and CCL5 have the ability to cluster and to activate eosinophilic granulocytes with the subsequent release of histamine and leukotrienes. The consequence is the induction of inflammatory cell adherence to the vessel wall [65]. Increased levels of CCL5 have been observed in patients with primary biliary cirrhosis [66]. It has been suggested that the expression of CCL3 and CCL5 are elevated in a time-dependent manner until seven days, indicating that these two molecules are crucial for the pathogenesis of biliary atresia in mice.

## 6. Non-Genetic Ambiguities

Several viruses have been associated with biliary atresia, although data are controversial. The viral consideration is linked to the Landing’s hypothesis. Professor Petersen’s group of Hannover, Germany that studied many samples found a viral etiology in only a minority of patients with biliary atresia [67,68]. A virus-related etiology was initially supported by Landing in 1974 [69]. There is an important point to make. There are cases of biliary atresia showing a close link to viruses. As indicated earlier, the pathogenesis is still elusive. In fact, it seems to be multifactorial. Other than a viral involvement with sporadic cases including reovirus, rotavirus, Epstein-Barr virus, and cytomegalovirus, a decreased Treg subset of the CD4 positive T lymphocytes [70,71], environmental factors, such as biliatresone [59,72,73,74,75], and ADP-ribosylation factor 6 (ARF6) have been actively investigated [76].

Moreover, there is a mouse model of biliary atresia which is induced by neonatal viral infection, and an immune response in patients affected with biliary atresia mimics the response to viral infections, i.e., they harbor an induction of Th_1_ cells and upregulation of Toll-like receptor (TLR) 3 and 7a as well as a high expression of Mx protein [74,77,78,79]. Although it can be argued that only a minority of patients with biliary atresia are positive for viruses, many cases have been tested for a few viruses only. Although the mentioned Professor Petersen’s group of Hannover, Germany, identified 27 out of 64 cases (42%) the sampling procedure may have been biased and viral DNA or RNA could be a secondary phenomenon due to a postnatal acquisition not involving the early stages of biliary atresia. Moreover, it needs to be strongly emphasized that the timing of virus testing is crucial. It means that when the biliary atresia specimens are tested, the liver tissue shows already an advanced stage of disease and virus clearing may act quite fast. Another critical aspect that needs to be considered is that the viral incidence in patients with and without simultaneous congenital disabilities was similar, which argues against a specific role of the viruses investigated in the non- syndromal form of biliary atresia [67,68]. 

## 7. Treatment

Although initially considered an extrahepatic pathology of the biliary tract, this disease is now correctly considered a pathology of the intra- and extrahepatic biliary tract. Currently and at least in several children’s hospitals, biliary atresia is surgically classified as type I (atresia of the common bile duct, 10%), type II (atresia of the hepatic duct, 2%), and type III (atresia of the porta hepatis, 88%), but also as embryonic with DPM and fetal type without DPM. Nevertheless, classifications may have limited value in biliary atresia. The porto-enterostomy ideated by the Japanese Morio Kasai, who introduced an operative procedure, called portoenterostomy, aimed to ‘‘correct the uncorrectable’’ [5]. It represents a breakthrough for the outcome of patients affected with biliary atresia. Although the risk of cholangitis may jeopardize the outcome, several factors have also been suggested. Among others, the presence of ductal plate malformation with the inadequacy of the intrahepatic biliary system to adequately drain the bile. Nowadays, Kasai’s series of patients are still alive, and their descendants are reported to have a healthy liver. In the case of liver portoenterostomy failure, a terrific milestone was the step of liver transplantation. Although the long-term survival in patients with biliary atresia and native liver improves continuously, most of these patients will need early or late liver transplantation. The success of the modern liver transplantation has reached an overall survival rate exceeding 90% in several studies [80,81,82,83,84,85]. Arafa et al. [86] evaluated 59 infants with neonatal cholestasis and separated them in two groups, i.e., biliary atresia group (*n* = 31) and non- biliary atresia group (*n* = 28). Interleukin-2 (IL-2) and interleukin-8 (IL-8) immunohistochemical staining in the portal cellular infiltrate of the liver was scored. These authors found that the mean value of IL-2 and IL-8 positive inflammatory cells was significantly higher in the biliary atresia than in the non- biliary atresia group and IL-2 correlated with IL-8 immunohistochemical staining in both biliary atresia and non- biliary atresia groups. In addition, both cytokines correlated significantly with inflammatory activity in liver biopsy in both groups while there was no significant correlation with the other studied parameters. In particular, the biliary atresia group showed that there was a trend of increased expression of IL-2 and IL-8 with increasing stage of fibrosis, which may have relevance as a marker of disease severity predicting the progression to liver fibrosis. The extent of biliary proliferation in liver biopsies from patients with biliary atresia at portoenterostomy has been associated with the postoperative prognosis and a Brazilian group [87] found that (cyto-)keratin 7 positivity (PCK7) ranged between 0.80% and 14.79% in their patients with biliary atresia. Patients who died or underwent transplantation had higher PCK7 than survivors with their native livers. Remarkably, the area under the receiver operating characteristic (ROC) curve for PCK7 in relation to the outcome was 0.845. The PCK7 was the only studied parameter associated with 1-year native liver survival, independently of age and fibrosis score. In a Japanese study, Obayashi et al. [88] reviewed 32 biliary atresia patients undergoing hepatoportoenterostomy operation from 1976 to 2016 with more than five portal canals in biopsy samples. These authors confirmed our original study with relevance of BD/PT ratio [8,88]. Obayashi et al. The study [8,88] suggests that the mean BDP ratio, i.e., the number of the bile ducts in portal canal ratio may be a better short-term prognostic indicator than PCK7. The mean BDP, PCK7, and PPCK7 increased with the patient’s age at the time of their Kasai operation. 

## 8. Conclusions

The crucial question that parents would ask the pediatrician is if their child can be cured. There is no significant uncertainty in saying that time will decide it. This feeling of hesitation is shared with several healthcare workers, not only pediatricians and pediatric surgeons. Indeed, biliary atresia can be cured satisfactorily with the split liver surgical procedures and/or liver transplantation, but there are a few questions that remain unanswered and precisely if some genes involved in biliary atresia may play a role in future diseases of their child. In addition, the complications related to the surgical procedures and the long-term anti-rejection therapy need to be discussed. To these and the many more questions that came out in the revision of our professional activity and literature data, the adoption of an intriguing approach toward biliary atresia remains crucial. Still, we need to stay humble because there are numerous questions that do not have answers. In the future, the institution of research chairs in biliary atresia by stakeholders should be strongly emphasized. The use of bioinformatics tools and new data science platforms will be decisive in addressing some investigations of this mysterious disease that has targeted newborns for the last four millennia.

## Figures and Tables

**Figure 1 diagnostics-12-00330-f001:**
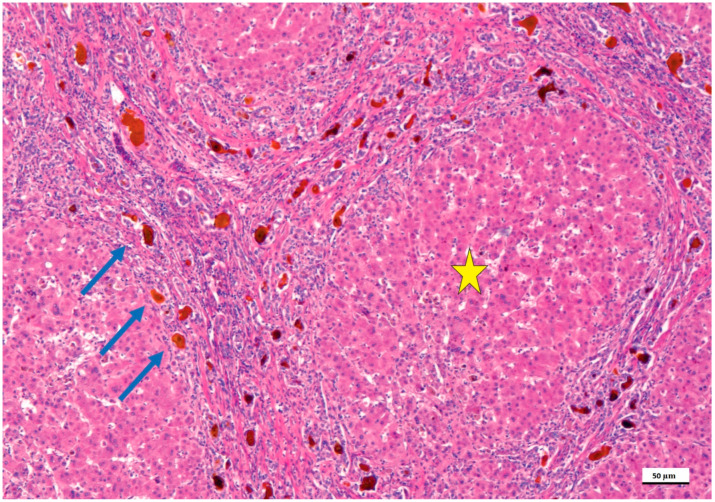
Biliary atresia in a child treated with the Kasai portoenterostomy, which failed. In this microphotograph, the formation of lobules and pseudo-lobules (yellow star) is apparent indicating stage IV fibrosis/cirrhosis. The portal tracts are fused highlighting the porto-portal and centro-portal bridges. In these bridges (blue arrows), the intrahepatic biliary tract shows biliary ductal proliferations and lumens filled with bile, which appears brown (hematoxylin and eosin staining, 50× as original magnification, bar: 50 μm).

**Figure 2 diagnostics-12-00330-f002:**
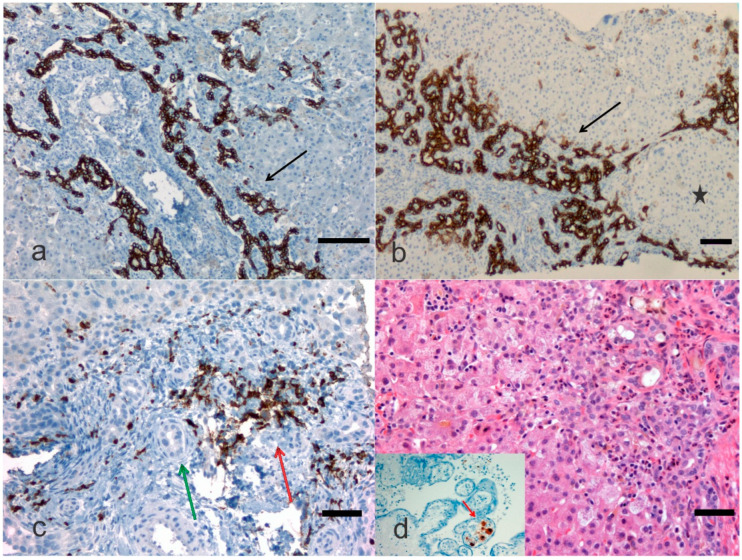
Biliary atresia with different clinical background. In this microphotograph panel, the bile ductular proliferations (arrows) are highlighted with a monoclonal antibody against cytokeratin or keratin 19 (**a**) and against cytokeratin or keratin 7 (**b**) by immunohistochemistry. Keratin-19 or cytokeratin-19 is a 40 kDa type I cytoskeletal protein that in humans is encoded by the *KRT19* gene. In (**b**) the bile ductular proliferation is particularly pronounced (arrow) and a pseudonodule is noted (black star). In (**c**), an antibody against myeloperoxidase is carried out in an early case of biliary atresia highlighting the neutrophilic inflammation (red arrow) close to bile ducts (green arrow). Staining is either by immunohistochemistry (this case) or enzyme cytochemistry. Myeloperoxidase is the most sensitive and specific stain for myeloid leukemias and granulocytic sarcoma, but it is also very useful to stain neutrophils, particularly when their identification may be difficult to catch when other cells confuse the histology. In (**d**) a child with biliary atresia and positive cytomegalovirus (CMV) serology, but negative CMV immunohistochemistry, may disclose CMV positive cells in the placenta ((**a**), Bar 50 μm, 100× original magnification; (**b**), Bar 50 μm, 50× original magnification; (**c**), Bar 10 μm, 200× original magnification; (**d**), Bar 10 μm, 200× original magnification, and inset, 200× original magnification).

## Data Availability

Publicly available data and articles were analyzed in this study. This data and articles can be found in PubMed, Scopus, and Google Scholar on “biliary atresia” of the last 23 years (1998–2021) using “biliary atresia”, “genetics“, and “diagnosis“ as key words.

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
