# Peer review of "Biliary Atresia: A Complex Hepatobiliary Disease with Variable Gene Involvement, Diagnostic Procedures, and Prognosis"

_diagnostics, 2022, doi:10.3390/diagnostics12020330_

Round 1
Reviewer 1 Report
This is my second review of paper authored by Sergi and Gilmour. The Authors responded to all my question and revised their manuscript.
There have been published many papers, including similar reviews, on biliary atresia (BA). However, the paper authored by Sergi and Gilmour, I found quite different and thus very interesting.
Besides of all, there is a some disproportion in the main text between pathological issues (chapter 2 and 3) and others; especially I regret that there is a scarce information regarding clinical issues on BA, e.g. clinical classification, timing of disease onset. On the other hand, most of published papers on BA raised these issues, thus I found this paper quite interesting and suitable for publication.
Reviewer 2 Report
This is a resubmitted manuscript (review of literature regarding biliary atresia). The authors performed mostly minor changes to the text.
- First, this is not the optimal search of literature that the authors have conducted. An optimal search in a systematic review and meta-analysis requires exploring at least 4 databases. Please see the link below: https://systematicreviewsjournal.biomedcentral.com/articles/10.1186/s13643-017-0644-y
- The authors stated that this review was based on the 30-years personal experience of patients with biliary atresia and literature search. Unfortunately, I do not see personal input of the author in this repost, or it is very weak, most of the paragraphs in this report are repetition of well-known facts from the literature.
- As it was stated in the title ‘’diagnostic procedures’’ I expected clinical diagnostic procedures / algorithms but the authors focused only on histology, genetics… In my opinion different clinical diagnostic modalities should be reviewed.
- This topic has been published many times and many similar reviews exist in medical literature. On the other hand there is also systematic review published on the same topic. What are new findings / conclusions from this report?
- This review is more like a chapter from a book and not a real review of literature. Actually this is a narrative review of very low scientific importance. The authors should review crucial clinical factors/symptoms and perform review on each variable (compare findings from large series of patients). The conclusions drawn from the authors are general and well known and have been reported many times.
Unfortunately I do not see any benefits from this report. All of this has been published several times in medical literature.
This manuscript is a resubmission of an earlier submission. The following is a list of the peer review reports and author responses from that submission.
Round 1
Reviewer 1 Report
Thank you very much for the opportunity to review this interesting overview on this complex topic.
However, there are several points I want to mention:
- The title could be more focused on the content of the manuscript.
- In the introduction you mention the Kasai procedure to be a palliative procedure as a bridge for transplant. Although, it is palliative indeed, the aim is native liver survival with improving numbers over the last decades.
- Please mention in methods the focus of your literature review/ search. I guess you did not chose "biliary atresia" exclusively
- Paragraph 4 "Pathological anatomy..." should include some information on the current role of liver biopsies in the diagnostic algorithm of BA and especially information on sensitivities of the mentioned items. Furthermore, differences of the histological features compared to other pediatric hepatopathies should be mentioned
- Paragraph 7 "Treatment". Multitude of studies have reported on prognostic factors for Kasai outcome (age, center load, BASM, CMV ...). Furthermore, infection should be changed to cholangitis.
- Paragraph 8: Split liver surgical procedures are liver transplantations. The problems of liver transplantation (donor pool, immunosuppressives) should be mentioned. Transplantation is not only a cure, it is a new disease with a better prognosis.
Reviewer 2 Report
The authors performed review of literature regarding biliary atresia.
- The authors stated that this review was based on the 30-years personal experience of patients with biliary atresia and literature search. Unfortunately, I do not see personal input of the author in this repost, or it is very week, most of the paragraphs in this report are repetition of well-known facts from the literature.
- This topic has been published many times and many similar reviews exist in medical literature. On the other hand there is also systematic review published on the same topic.
- This review is more like a chapter from a book and not a real review of literature. The authors should review crucial clinical factors/symptoms and perform review on each (compare findings from large series of patients).
- Also, the most important part - clinical presentation and clinical diagnosis of biliary atresia was not even mentioned.
- The conclusions drawn from the authors are general and well known and have been reported many times.
- Most importantly, I do not see any benefit for the readers from this review because most of the presented information can be found in any basic pediatric book.
- The title of this Journal is ‘’Diagnostics’’ and there is not even a word about clinical diagnostics of biliary atresia.
Reviewer 3 Report
Sergi and Gilmour provided an overview of biliary atresia. As an article is based mainly on the 30-years personal experience of patients with biliary atresia, some aspects/issues were omitted by the Authors.
(1) 2 types of BA - does this division still exist?
Symptoms could appear after the jaundice-free period. In this type of BA, called isolated form, congenital anomalies are not observed. The second (fetal) type is associated with the biliary atresia splenic malformation (BASM) syndrome.
(2) The article was submitted to ,,Diagnostics''. Thus, Authors could more comprehensively provide an overview of liver biopsy in diagnosis and prognosis of BA.
(a) Liver biopsy is a helpful tool for evaluation of microscopical changes in the liver, especially ductular proliferation (DP), ductal plate malformation (DPM), fibrosis and cholestasis. However, these features, especially DP, and DPM, are not always distinct using routine staining. Visualization of cytokeratins and neural cell adhesion molecules in the immature ductal cells by means of the immunohistochemical method can be a useful tool for the microscopic examination of the immature biliary structures in the liver.
(b) There is a lot of studies in biliary atresia involving liver biopsy parameters as prognostic factors. Please, provide a short overview.
(3). The pathogenesis of the disease is still unclear. It is a multi-factorial disease. Some factors were omitted: ADP-ribosylation factor 6 (ARF6), environmental factors such as biliary toxin biliatresone, infectious causes including reovirus, rotavirus, cytomegalovirus
(CMV) and Epstein-Barr virus (EBV), and immune dysregulation, e.g. decreased Treg subset of CD4+ function.
(4) Elevated IL-8 in the serum or liver of patients with BA could be a useful marker of disease severity, could predict the progression to liver fibrosis, and could even serve as a useful biomarker for BA diagnosis.
In conclusion, I recommend to publish the artcile after major revison.